# OpenReview forum: "When Search Goes Wrong: Red-Teaming Web-Augmented Large Language Models"
_ICML.cc/2026/Conference — ICML 2026 regular_

### Official Review · Reviewer_wJKB · 2026-03-12

**Soundness:** 2
**Presentation:** 2
**Significance:** 2
**Originality:** 2
**Overall Recommendation:** 3
**Confidence:** 4

**Summary:**

This paper introduces CREST-Search, a red-teaming framework designed to evaluate the safety of LLMs augmented with web search (e.g., GPT-4o-search, Gemini-2.0-flash-search). The central observation is that search-augmented LLMs face a unique "citation risk": the model may retrieve and cite harmful or low-credibility webpages even when its generated response is benign. The framework proposes three adversarial query generation strategies—keyword injection, exaggeration, and role play—to craft seemingly innocuous queries that induce unsafe citations. It incorporates a judgment-feedback iterative refinement loop operating under black-box constraints and constructs a specialized dataset (WebSearch-Harm, ~1,527 instances) for fine-tuning a red-teaming model via SFT. Experiments across four commercial systems report an 80.5% risk detection rate, far exceeding baselines (5.2–11.6%), with 74.7% of detected risks being citation-only (the response itself is benign). The paper claims this is a "pioneering" framework for red-teaming search-augmented LLMs.

**Compliance With Llm Reviewing Policy:**

Affirmed.

**Final Justification:**

I appreciate the authors' more detailed second response and the additional persona-based baseline (12.3% vs 80.5%).

However, my core concern remains. The authors argue the strategies are not "relabeling known techniques" by emphasizing differences in attacker capability and attack goal. Yet the fundamental mechanisms—steering retrieval via keyword manipulation, reshaping query semantics, and adopting personas to bypass filters—are well-established ideas. That they require adaptation to work in the citation-risk setting is expected for any technique applied to a new domain; this does not elevate the contribution to methodological novelty. The SafeSearch and persona-based comparisons both suffer from the same asymmetry issue I raised earlier: these methods were not designed for citation risk, so their underperformance is unsurprising rather than evidence of novelty.

Regarding Q2, I accept the authors' apology and the ICML guideline reference. My frustration was that key data points central to my own questions were absent from my rebuttal, not merely that a shared question was answered once.

I maintain my score of 3 (Weak Reject). The paper addresses an important problem, but the technical novelty of the attack strategies remains incremental, and the paper would benefit from a more honest positioning as "known strategies effectively adapted and validated in an underexplored setting" rather than claiming novel strategies.

**Key Questions For Authors:**

check the weakness

**Limitations:**

No. The authors have not adequately discussed the limitations of their work. Key undiscussed limitations include: (1) reliance on automated moderation tools without human validation and the associated false positive risk; (2) non-reproducibility of experiments on evolving commercial APIs; (3) the narrow harm taxonomy that excludes misinformation/disinformation—arguably the most relevant harm category for search systems; (4) circularity in dataset construction (optimizing and evaluating on the same model family); (5) **complete omission of Agent-SafetyBench from the related work**, which has already addressed the broader problem of LLM agent safety including search tools; (6) omission of SafeSearch as an experimental baseline despite citing it.

**Strengths And Weaknesses:**

**Strengths:**

1. **Timely and practically relevant problem (Significance).** As more commercial LLM products integrate real-time web search (ChatGPT Search, Gemini with Google Search), the risk of users being exposed to harmful content *through cited links*—rather than through the model's generated text—is a genuine and growing concern. The paper correctly identifies that this threat surface differs from standard LLM safety evaluation.

2. **Strong reported empirical performance.** The gap between CREST-Search (80.5%) and the best baseline (HarmBench at 11.6%) is large. The framework also achieves the lowest query toxicity (23.6%) among all methods, demonstrating that the adversarial queries are not trivially offensive. Transferability across four commercial models (two GPT-series, two Gemini-series) adds breadth.

3. **Useful risk taxonomy.** The three-level risk classification (response risk, citation risk, combined risk) is a useful conceptual contribution. The finding that 74.7% of detected risks are citation-only is noteworthy and highlights a genuine blind spot in current evaluation practices.

4. **Comprehensive experimental design.** Evaluating across 5 categories × 3 strategies × 4 target models with 1,500 test cases per run (repeated 3 times), using multiple moderation tools (OpenAI Moderation API, LlamaGuard), demonstrates systematic effort.

5. **Low cost and efficiency.** The ablation study showing the fine-tuned model achieves superior performance at only $1.50 per 100 queries and 49.6 seconds per query is practically useful.

**Weaknesses:**

1. **Critical missing related work: Agent-SafetyBench (Zhang et al., 2024) (Originality, Presentation).** This is perhaps the most significant omission. Agent-SafetyBench systematically evaluates the safety of LLM agents—including agents that use web search as a tool—across 349 environments and 2,000 test cases. It explicitly addresses safety risks arising from external tool interactions (including search), categorizes risk sources across different stages (user input, external environment, agent behavior), and evaluates multiple commercial LLMs under black-box conditions. This work directly subsumes a substantial portion of the problem space that CREST-Search claims to be "pioneering." The authors' claim in the abstract that they propose "a pioneering red-teaming framework for LLMs with web search" is significantly undermined by the existence of Agent-SafetyBench, which already provided a broader framework encompassing search-tool safety within the LLM agent paradigm. The paper's Related Work (§2) covers three areas—LLMs enhanced by web search, adversarial attacks, and red-teaming LLMs—but entirely omits the critical direction of **LLM agent safety**, where Agent-SafetyBench is a representative and well-known work. This omission is not merely a citation gap; it reflects a fundamental failure to position the work within the correct and most relevant body of literature.

2. **SafeSearch (Dong et al., 2025) cited but not compared (Soundness).** The paper cites SafeSearch in the introduction (line 235) and references (line 1431–1434), acknowledging it as a concurrent work on "automated red-teaming for the safety of LLM-based search agents." Yet SafeSearch is conspicuously absent from the baseline comparisons (§5.1). Given that SafeSearch directly targets the exact same problem—red-teaming LLM-based search systems—its omission from experiments is a critical gap that makes the reported performance improvements less convincing. Without this comparison, it is impossible to know whether CREST-Search's advantage comes from genuinely better methodology or simply from targeting the right attack surface (which SafeSearch also does).

3. **Unfair baseline comparison (Soundness).** All four baselines (UAT, RED-EVAL, DangerousQA, HarmBench) were designed for standalone LLMs and were never intended to target citation risks. The 80.5% vs. 11.6% performance gap is therefore largely expected—it reflects that baselines target a fundamentally different attack surface, not that CREST-Search's methodology is superior. A fair evaluation would require: (a) comparing against SafeSearch and Agent-SafetyBench or similar agent-safety methods; (b) adapting existing baselines to explicitly target citation risk; (c) or at minimum, acknowledging that the comparison is inherently asymmetric.

4. **Questionable novelty of attack strategies (Originality).** The three strategies—keyword injection, exaggeration, and role play—are adaptations of well-established techniques: SEO keyword stuffing/poisoning, query manipulation via hyperbole, and persona-based jailbreaking (Deshpande et al., 2023; and many others). The paper does not provide sufficient technical novelty beyond applying these known ideas to a new setting. The iterative refinement pipeline follows the established pattern from MART (Ge et al., 2023), and the fine-tuning is standard SFT. When prior work (Agent-SafetyBench, SafeSearch) has already addressed the same setting, even the "application novelty" is diminished.

---

> ### Author Rebuttal · Authors · 2026-03-31
>
> We thank the reviewer for the detailed feedback and valuable comments.
>
> ### Q1. Clarification on Agent-SafetyBench
> We agree that Agent-SafetyBench is a valuable benchmark for evaluating LLM agent safety. However, we respectfully clarify that our work is not subsumed by Agent-SafetyBench, as the two address different problems:
> - Agent-SafetyBench: safety in tool-augmented agents under simulated environments
> - Ours: safety risks arising from open-web retrieval and citation pipelines
>
> To the best of our understanding, Agent-SafetyBench does not consider open-domain retrieval setting based on real web content. It focuses on evaluating the safety of LLM agents in simulated, closed-world environments with predefined tools.
>
> While it considers tool usage broadly, including retrieval-like operations, it **does not involve real-world web search**. Therefore, the comments *“encompassing search-tool safety”* may mismatch what is implemented in Agent-SafetyBench. Risks such as citation risk induced by external web content are not captured in Agent-SafetyBench.
>
>
> In contrast, our work specifically targets **search-enabled LLMs operating over the open web**, where information is dynamically retrieved from external sources. We respectfully believe that the claim that our work is *“significantly undermined”* by Agent-SafetyBench may not fully reflect the difference in problem settings, as the two works address fundamentally different types of risks.
>
> We clarify that our contribution lies in identifying and systematically evaluating **citation risk in search-enabled LLMs**. As also noted by Reviewer-LKaP, *“Citation risk as a category while obvious hasn't received a lot of attention in the academic literature.”*
>
> We will include Agent-SafetyBench and clearly describe our contribution to avoid potential overstatement.
>
>
> ### Q2. Baseline comparison
>
> Following the reviewer’s suggestion, we add a comparison with SafeSearch. The experimental results show that our method achieves stronger performance. The detailed experimental results and discussion of the differences between SafeSearch and our framework, are provided in *Reviewer-LKaP, Q3*.
>
> In addition, the results show that directly applying existing baselines(UAT, RED-EVAL, DangerousQA, HarmBench) leads to limited effectiveness in triggering citation-related risks. This highlights the significance of the research problem and the effectiveness of our proposed framework.
>
>
> ### Q3. Novelty of attack strategies and overall framework
>
> **The novelty of attack strategies.** We refer the reviewer to *Reviewer-LKaP, Q1*, where we provide a detailed clarification on the novelty of our attack strategies. This design has also been positively recognized by *Reviewer-GHhD, Strength 2*.
>
> **Clarification on iterative refinement.** We acknowledge that iterative refinement has been explored in prior work (e.g., MART). However, our framework operates in a different setting and requires a corresponding shift in both design and feedback signals.
>
> **Clarification on overlap with prior work.** We respectfully refer the reviewer to our discussions in *Q1* (Agent-SafetyBench) and *Reviewer-LKaP, Q3* (SafeSearch), where we clarify that these works address different settings.
>
>
> ### Q4. Discussion of limitations
>
> We appreciate the reviewer’s suggestions and will incorporate a more comprehensive and explicit discussion of these limitations in the final version.
>
> **(1) Reliance on automated moderation tools.**
> We refer the reviewer to our response to *Reviewer-GHhD, Q4*, where we provide a detailed discussion on the reliability and robustness of using automated moderation tools in our evaluation framework.
>
> **(2) Reproducibility with commercial APIs.**
> Plese refer to *Review-GHhD，Q2*, where we provide the API configurations used in our experiments for reproducibility. We also conduct repeated runs to verify the stability of our results in the draft *Section 5.1, Configuration*.
>
> **(3) Scope of harm taxonomy.**
> We clarify that our work focuses on harmful content, rather than specifically targeting misinformation/disinformation. Our taxonomy follows commonly used categories in prior safety research[[1]](https://arxiv.org/pdf/2312.06674) (e.g., harassment, illicit, self-harm), which are widely adopted in existing moderation frameworks.
>
> **(4) Potential circularity in dataset construction.**
> We understand the concern regarding potential bias when using similar model families in attack generation and evaluation. In our setting, we evaluate across multiple victim models and adopt a black-box framework, reducing the risk of overfitting to a specific model. This suggests that the observed effects are not limited to a single model family.
>
> We will incorporate more detailed analyses discussed above in the revised version, and carefully improve the presentation.

---

> > ### Author Rebuttal · Reviewer_wJKB · 2026-04-02
> >
> > I thank the authors for their response. However, my core concerns remain largely unresolved.
> >
> > 1. Novelty of attack strategies. The authors argue that keyword injection, exaggeration, and role play differ from their known counterparts (SEO poisoning, query amplification, jailbreaking) because they "operate at the input level" or "focus on inducing unsafe citations." This is essentially application novelty—adapting known techniques to a new setting. Reviewer LKaP independently reached the same conclusion: "I see them as three known strategies adapted to a new setting." The authors themselves acknowledge the need to "revise the presentation to avoid overclaiming," which implicitly concedes this point. Relabeling known techniques for a new context does not constitute the "three novel adversarial search query generation strategies" claimed in the abstract.
> >
> > 2. Over-reliance on cross-referencing other reviewers' responses. The rebuttal to my review repeatedly uses "refer the reviewer to Reviewer-LKaP Q1/Q3" and "refer the reviewer to Reviewer-GHhD Q4" instead of directly addressing my questions. This is not an acceptable rebuttal practice. Each reviewer's concerns deserve a self-contained response. I should not have to piece together answers from responses written for other reviewers with different questions and contexts.

---

> > > ### Author Response · Authors · 2026-04-07
> > >
> > > ### Q1. Novelty of attack strategies
> > >
> > > We thank the reviewer for the concern. We clarify that our approach is not a direct application of known techniques to a new setting. Additionally, our statement about *”revising the presentation to avoid overclaiming“* does not imply a lack of novelty, but rather aims to more precisely position our contribution and novelty. The novelty of the attack strategies is shown as follows:
> > >
> > > **(1) Keyword injection vs. SEO poisoning.**
> > > *First*, the attacker's capability is fundamentally different. SEO poisoning assumes the ability to manipulate or inject content into external webpages and relies on those manipulated pages being retrieved. In contrast, our framework operates at the query level and does not require any control over external webpages.
> > > *Second*, the technique differs. SEO poisoning optimizes webpage content to influence ranking. Keyword injection is designed to steer retrieval behavior through query formulation under input-only constraints.
> > > *Third*, our experiments show that these methods do not directly transfer to this scenario. For example, SafeSearch evaluates under an offline setting, where the attacker can manipulate retrieved pages. Our experiments show that such approaches perform poorly under a purely online, black-box scenario. We argue that our setting more closely reflects the real-world scenario.
> > >
> > > |Method|Detection Rate|
> > > |-|-|
> > > |SafeSearch|38.3%|
> > > |CREST-Search|80.5%|
> > >
> > > **(2) Exaggeration vs. query amplification.**
> > > Exaggeration does not simply increase query strength. Instead, it subtly reshapes the semantic emphasis of the query while maintaining plausibility. This is crucial in the LLM-search setting, where overly explicit or aggressive queries are more likely to be filtered or blocked. Our strategy is designed to keep queries natural and benign-looking while still influencing retrieval outcomes, which differs from amplification methods that rely on increasing explicit signal strength.
> > >
> > > **(3) Role play vs. jailbreak prompting.**
> > > *First*, the goal is fundamentally different. Jailbreaking aims to elicit unsafe generation from the model, whereas our role-play strategy is designed to influence retrieval and citation behavior without triggering unsafe responses.
> > > *Second*, our experiments in the paper(Table 1) show that existing jailbreak/red-teaming baselines(UAT, READ-EVAL, etc.) are ineffective in this setting. This is not because they are blocked or filtered(69.4% of baseline queries receive valid responses), but because their design paradigm does not transfer to this scenario.
> > > *Third*, we also add related work mentioned by the reviewer(Deshpande et al., 2023)[1], which assigns LLM with different personas to increase the toxicity of the generated response for comparison. The experimental results are shown below. Our method achieves a higher detection rate.
> > >
> > > |Method|Detection Rate|
> > > |-|-|
> > > |Persona-based|12.3%|
> > > |CREST-Search|80.5%|
> > >
> > > Overall, our attack strategies are not relabeling known techniques for a new context. The results also demonstrate that directly applying known techniques to this setting is not effective. This highlights the necessity and effectiveness of our proposed novel strategies, which are designed under a different paradigm. Additionally, our contribution also lies in **a unified red-teaming framework** for evaluating citation risk in search-enabled LLMs, which has not been addressed in prior work. After our clarification, *Reviewer LKaP* also acknowledged the novelty of our approach.
> > >
> > > We will revise the paper to better distinguish our contribution and novelty from prior techniques.
> > >
> > > [1] Deshpande A, Murahari V, Rajpurohit T, et al. Toxicity in chatgpt: Analyzing persona-assigned language models[C]//Findings of the association for computational linguistics: EMNLP 2023. 2023: 1236-1270.
> > >
> > > ### Q2. Self-contained responses to reviewer concerns
> > >
> > > We sincerely apologize for the additional burden introduced by the cross-referencing. We adopted this approach to ensure that all concerns were addressed within the limited rebuttal space. Therefore, we prioritized directly responding to most of your specific questions while referring to responses to similar questions raised by other reviewers.
> > >
> > > The ICML guidelines also recommend that it is fine to point a reviewer to the response written for a different review. In the official email, it states that "if two reviewers ask the same question, the author can answer it in the response to one reviewer and then ask the other reviewer to find the answer there."

---

### Official Review · Reviewer_GHhD · 2026-03-12

**Soundness:** 2
**Presentation:** 2
**Significance:** 3
**Originality:** 3
**Overall Recommendation:** 3
**Confidence:** 4

**Summary:**

The paper introduces CREST-Search, a novel red-teaming framework designed specifically to evaluate the safety of Large Language Models (LLMs) augmented with web search capabilities. The authors identify a critical gap in current safety evaluations: while traditional red-teaming focuses on the model's direct text generation (response risk), web-augmented LLMs introduce "citation risk" by retrieving and citing harmful or low-credibility content from the open internet. To expose these vulnerabilities under black-box constraints, CREST-Search employs three specialized attack strategies: keyword injection, exaggeration, and role-play. The framework uses an automated pipeline for adversarial query generation, execution, and judgment-guided refinement. Furthermore, the authors curate a specialized dataset named WebSearch-Harm to fine-tune a dedicated red-teaming model, enabling highly efficient, single-step adversarial query generation. Evaluating against multiple commercial models, the framework achieves a high risk detection rate of 80.5\%, demonstrating that current web-augmented systems are highly susceptible to unsafe citations even when the model's direct response appears benign.

**Compliance With Llm Reviewing Policy:**

Affirmed.

**Final Justification:**

Although the idea is decent and I agree with the authors that the approach is effective, the presentation quality of this paper has not yet reached the standard required for publication. Presentation also falls within the four dimensions against which the quality of this work is assessed. Considering that the methodology is reasonable and effective enough, I decide to slightly raise my review score to 3. If the authors seriously improve the clarity and writing quality of the paper, I believe there is still a good chance that it will be accepted by a top-tier conference.

**Key Questions For Authors:**

I will consider changing my ratings based on the authors' response.

1. How robust are the automated moderation tools when applied to raw web page content? Modern web pages usually contain complex components like ads, navigation elements and fragmented text. The accuracy of the detectors on the retrieved web content should be validated. The authors can demonstrate a specific example of deploying CREST-Search framework if convenient.

2. Why are other commercial web search tools not considered as alternative (e.g. Claude and Perplexity)? Is it possible to evaluate CREST-Search on an open-source RAG or search agent (models with MCP) architecture? Despite your justification for using commercial models in Appendix A.1 is practical, the experiments are somewhat not sufficient enough only considering Gemini and OpenAI models.

**Limitations:**

yes

**Strengths And Weaknesses:**

Strengths:

1. As major AI labs aggressively integrate web search into their flagship models, identifying and mitigating the safety risks of the retrieval-citation pipeline is an urgent problem. By demonstrating that 74.7\% of total uncovered risks involve safe-looking text that cites harmful sources, the paper highlights a massive blind spot in current AI safety protocols. The proposed fine-tuning approach which reduces costs to \$1.50 per 100 queries provides clear practical utility for industry practitioners.

2. Shifting the red-teaming objective from eliciting toxic generation to inducing unsafe external citations is a highly original and necessary perspective for search-augmented agents. The specialized search query generation strategies (especially keyword injection resembling SEO bait) creatively exploit the specific mechanics of search engine retrieval.

Weaknesses:

1. The authors could have arranged the structure of the paper better. Many figures and tables in the paper are too small and hard to view (especially Figure 1, Figure 5). Also, in Section 4.1, the authors at least should have briefly described the harmful content category instead of putting these content into Appendix A.3 without in-text citation.

2. The experiments of this work are way too much from rigorous.

(1) The middle and the right subfigure of Figure 5 are totally the same. Have the authors double-checked the content of the paper?

(2) While the paper mentions that queries were executed without any system prompt settings, it lacks minor reproducibility details regarding the hyperparameter settings (e.g., temperature, top-p) used when querying the target victim APIs.

(3) The authors also don't clarify how they extract these URLs and fetch the raw content of web pages under a purely black-box API or web interface stably and automatically at least in order to evaluate if the content in cited websites is unsafe.

3. Regardless of the flaws, the primary weakness of the idea lies in the evaluation metric for attack success. The framework relies entirely on automated moderation tools (OpenAI moderation and Llama Guard) to evaluate the safety of the cited external webpages. Web pages are structurally complex and noisier than standard LLM text outputs. The lack of human validation to confirm the false positive/negative rates of these detectors on raw web data limits the absolute reliability of the reported 80.5\% detection rate. Furthermore, comparing CREST-Search against baselines like HarmBench and UAT is somewhat unfair, as those baselines were never designed to manipulate retrieval mechanisms.

---

> ### Author Rebuttal · Authors · 2026-03-31
>
> Thanks for the reviewer's valuable feedback and insightful comments; we address these concerns as follows.
>
> ### Q1. Presentation
> We will thoroughly proofread the paper and refine the overall structure in the revision:
> (i) improve the readability of figures and tables(enlarge text size and correct the duplication error in Figure 5)
> (ii) reorganize brief descriptions of harmful content categories in Section 4.1.
>
> ### Q2. Reproducibility details
>
> We provide the API hyperparameters used in the target victim API and CREST-Search. We use the default settings of these models.
>
> |Model|Temperature|Top-p|
> |-|-|-|
> |GPT-4o|1.0|1.0|
> |GPT-4o-search-preview|1.0|1.0|
> |GPT-4o-mini-search-preview|1.0|1.0|
> |Gemini-2.0-flash-search|1.0|0.95|
> |Gemini-2.5-flash-search|1.0|0.95|
>
> In the revision, we will provide a more comprehensive details to enhance the reproducibilty.
>
>
> ### Q3. Clarification of webpage extraction
>
> We apologize for the missing details regarding the extraction process. In our framework, we use the following pipeline to process web content under black-box constraints:
> - URL Extraction: We extract URLs directly from the citation metadata provided in the model's response.
> - Content Fetching: We use a headless browser to fully render each page, which allows us to handle dynamic content.
> - Noise Reduction: To ensure stable evaluation, we extract only the main visible text and filter out irrelevant elements like advertisements, navigation bars, and scripts.
> - Safety Assessment: This cleaned text is then passed to moderation tools to identify potential risks.
>
>
> We will provide a more detailed description of this process in the revision to improve reproducibility and clarity.
>
> ### Q4. Reliability of moderation tools
> As mentioned in Q3, we design a preprocessing pipeline to reduce noise before evaluation. Additionally, we adopt two independent moderation systems (OpenAI Moderation API and Llama Guard) for cross-validating to improve robustness and reduce potential bias in detection.
>
> We would like to clarify that the use of automated moderation tools for red-teaming or safety evaluation is common in prior peer-reviewed works[1]. These tools (OpenAI Moderation API and Llama Guard) have been validated in their respective technical reports and are widely adopted in the literature for detecting harmful content[2],[3]. Using them allows scalable evaluation across a large number of test cases.
>
> To further validate the reliability of automated moderation tools, we conduct a small-scale human evaluation on a randomly sampled subset of cases. The results show a high level of agreement between human annotations and automated tools, indicating that the tool-based evaluation is reasonably reliable for our task. Finally, since all methods are evaluated using the same tools, the comparison remains fair even if the tools are slightly imperfect.
>
> |Metric|Value|
> |-|-|
> |Sample size|100|
> |Agreement(Human vs. Tools)|91%|
> |False Positive Rate (Tools)|5%|
> |False Negative Rate (Tools)|4%|
>
>
> We will include an illustrative example in the revision to demonstrate the full pipeline of CREST-Search.
>
> [1] Li L, Dong B, Wang R, et al. Salad-bench: A hierarchical and comprehensive safety benchmark for large language models[C]//Findings of the Association for Computational Linguistics: ACL 2024. 2024: 3923-3954.
>
> [2] Llama Guard 3 Model Card, https://github.com/meta-llama/PurpleLlama/blob/main/Llama-Guard3/8B/MODEL_CARD.md
>
> [3] Moderation API, https://openai.com/index/upgrading-the-moderation-api-with-our-new-multimodal-moderation-model/
>
>
> ### Q5. Other alternative victim models
>
> We conducted additional experiments on another commercial system Perplexity AI. We report the results on "harassment" harmful content category across 3 attack strategies below. The results show that CREST-Search achieves high risk detection rates across different systems, indicating good transferability.
>
> |Attack Strategy|Risk Detection Rate|
> |-|-|
> |Key Words Injection|84.8%|
> |Exaggeration| 82.1%|
> |Role Play|83.6%|
>
> While we do not include experiments on open-source systems, our method is expected to remain effective. Because, open-source LLM-based pipelines generally exhibit weaker safety alignment compared to commercial systems. We will include a more comprehensive evaluation in the final version.
>
> We will incorporate more detailed experiments and analyses discussed above in the revised version, and carefully improve the presentation.

---

> > ### Author Rebuttal · Reviewer_GHhD · 2026-04-02
> >
> > Thanks to the authors for addressing much of my concerns. However, I think some of them (Weakness 1, 2) cannot be fully resolved in this short rebuttal period. Although the idea is decent, the quality of this paper has not yet reached the standard required for publication. Therefore I decide to maintain my original view and keep my review score. If the authors seriously improve the clarity and writing quality of the paper, I believe there is still a good chance that it will be accepted by a top-tier conference.

---

> > > ### Author Response · Authors · 2026-04-07
> > >
> > > ### Solution for Weakness 1,2
> > >
> > > We thank the reviewer for the question. We apologize that our initial response did not fully address all of your concerns. Below, we provide clarifications for the remaining points:
> > >
> > > **(1) Presentation issue.** The issues are primarily presentation-related rather than methodological. We provide concrete solutions to address these concerns below. Specifically:
> > >
> > > - The duplication error in Figure 5 has been identified and corrected. Figure 5(middle) presents the impact of refinement rounds on API cost.
> > > - We enlarged figures and tables to improve readability.
> > > - The harmful content categories are already referenced in the original draft at the end of Section 5.2. In the revision, we will add a brief in-text description of these categories and make the reference to Appendix A.3 more explicit.
> > >
> > > We have also incorporated these updates into our local revision to improve the clarity and completeness of the manuscript.
> > >
> > > **(2) Reproducibility details.** Following the reviewer’s suggestion, we provide the API hyperparameters used across all models in our previous response to *Q2*. These parameters are fixed for all experiments to ensure consistency.
> > >
> > > **(3) Webpage extraction and evaluation pipeline.** We clarify that our extraction process is fully automated and designed for stable evaluation under black-box constraints. It enables CREST-Search to evaluate cited websites' safety at scale while maintaining high fidelity to the content actually seen by the users. Specifically, we:
> > > - Extract citation URLs from the structured metadata provided by the model APIs.
> > > - We employ a Selenium-driven headless Chrome browser to automatically and stably extract web content. This allows us to:
> > > (i) Execute JavaScript and capture content that appears only after full page rendering.
> > > (ii) Extract visible text. By accessing the rendered `body.text` attribute, we filter out non-visible components such as HTML tags, meta scripts, and hidden advertisement slots.
> > > (iii) Noise reduction. We apply post-extraction cleaning to remove fragmented text and short navigational elements, ensuring that moderation tools process the most relevant textual content for risk assessment.
> > >
> > > Overall, we believe that these issues do not significantly affect the effectiveness of our approach. We will incorporate these improvements in the revision to make it clearer. We thank the reviewer again for further questions, which help us improve the overall assessment of our work.

---

### Official Review · Reviewer_LKaP · 2026-03-12

**Soundness:** 2
**Presentation:** 2
**Significance:** 3
**Originality:** 3
**Overall Recommendation:** 4
**Confidence:** 4

**Summary:**

This paper introduces CREST-Search. This is a new red-teaming framework for LLMs that use web search. The paper targets the retrieval citation pipeline. The authors design three attack strategies: keyword injection, exaggeration, and role play. These generate benign-looking queries that push the LLM to cite harmful webpages. The framework refines queries iteratively and trains a specialized red-teaming model on a curated dataset called WebSearch-Harm. The authors test their approach on four commercial LLMs and report 80.5% risk detection rate. \~70% of the detected risks are citation-only i.e. the response looks safe, but the cited pages contain harmful content. The main contribution of this paper is showing that citation risk exists as a category and that current red-teaming methods miss to identify these.

**Compliance With Llm Reviewing Policy:**

Affirmed.

**Final Justification:**

All proposed changes look reasonable and doable within the timeline and thus I have raised my score.

**Key Questions For Authors:**

See weakness above

**Limitations:**

yes

**Strengths And Weaknesses:**

## Strengths

- Citation risk as a category while obvious hasn't received a lot of attention in the academic literature. I found the framing of the risk convincing. Many failure cases come from harmful citations while the response itself stays benign. This shows that response-level safety filters do not protect users from search-enabled systems. The community should pay attention to this.

- The threat model is realistic. The authors treat commercial search LLMs as black boxes. They only submit queries and inspect outputs. This matches what third-party auditors face in practice. The attacks transfer across GPT and Gemini model families, which supports the practical relevance.

- The paper does a good job in separating the risk by their risk level: response risk, citation risk, and combined risk. It reports results for each one. This helps to understand where defenses need to go in the pipeline.

## Weaknesses

- The three attack strategies are not novel individually. Keyword injection is similar to SEO poisoning. Exaggeration is a form of query amplification. Role play is a standard jailbreaking technique. The novelty is in applying them to the citation pipeline. But the abstract is overclaiming them "three novel adversarial search query generation strategies." I see them as three known strategies adapted to a new setting. I would suggest tightening the language around this.

- The related work section is relatively thin and misses some relevant important recent papers in the red-teaming literature. I am sure there are others too.
	- [Operationalizing a Threat Model for Red-Teaming LLMs](https://arxiv.org/abs/2407.14937) (Verma et al 2025): This paper builds a framework for threat models in red-teaming. The threat model in Section 3.2 fits naturally into that taxonomy. Verma et al. separate different adversary capabilities and goals. This maps directly to the black-box constraint and citation-manipulation objective in this paper. I would suggest the authors to contextualize their work in this framework to help practitioners situate their contributions in the broader red-teaming attack and defenses landscape. This probably fits under the Infusion Attack category.
	- [Rainbow Teaming: Open-Ended Generation of Diverse Adversarial Prompts](https://arxiv.org/abs/2402.16822) (Samvelyan et al 2024): This paper generates adversarial prompts in an open-ended way. How does the fixed-strategy approach in CREST-Search compare to open-ended methods?
	- [Lessons From Red Teaming 100 Generative AI Products](https://arxiv.org/abs/2501.07238): This paper gives practical lessons from large-scale red-teaming. They apply directly here.
	- [RedTeamCUA: Realistic Adversarial Testing of Computer-Use Agents in Hybrid Web-OS Environments](https://arxiv.org/abs/2505.21936): This paper red-teams agents in web environments. It overlaps with the web-search setting.

- Maybe I am missing something but it seems to me that the baseline comparison is not fair. UAT, RED-EVAL, DangerousQA, and HarmBench target standalone LLMs. None of the existing techniques look at citation risk. The authors compare CREST-Search against methods that ignore citations and report an large gap. This overstates the contribution. It is more like an ablation that shows search-specific strategies matter. The authors cite SafeSearch but do not compare against it. SafeSearch targets search-augmented LLM safety directly and appears to be a better baseline to compare against.

- The defenses proposed are thin and not discussed much in the paper.

---

> ### Author Rebuttal · Authors · 2026-03-31
>
> Thanks for the reviewer's valuable feedback and insightful comments; we address these concerns as follows.
>
> ### Q1. Clarification on the novelty of the proposed attack strategies.
>
> We'd like to clarify that our strategy names may appear similar to existing techniques, but they differ in both the **attack settings** and the **operational constraints**, requiring specific adaptations to make them effective in the LLM-search scenario. As *Reviewer-GHhD* noted, our approach "creatively exploits the specific mechanics of search engine retrieval".
>
> - **Keyword injection vs. SEO poisoning.**
>   SEO poisoning involves manipulating external web content (e.g., creating or modifying webpages to influence search rankings). In contrast, our method operates purely at the **input level**, without altering or modifying existing web pages or injecting new web pages. We adapt the intuition of keyword influence to LLM-based search, where retrieval is coupled with reasoning. Our method further incorporates adversarial query construction to influence both retrieval and downstream reasoning, which differs from traditional keyword-based search manipulation.
> - **Exaggeration vs. query amplification.**
>   Compared to query amplification, exaggeration does not simply increase query strength. Instead, it is designed to subtly reshape the semantic emphasis of the query while seeming plausible. Our goal is to keep the input query natural to bypass safety filters while still influencing retrieval outcomes. This is particularly important in the LLM-search setting. Overly explicit or aggressive queries are more likely to be blocked.
> - **Role play vs. jailbreak techniques.**
> Existing jailbreak prompts often fail in search-enabled systems because they are too toxic and get blocked. We adapt "role play" into a **search-oriented context**, focusing on inducing unsafe citations rather than direct harmful generation.
>
> Importantly, our contribution is not limited to individual strategies, but also lies in the **unified red-teaming framework for search-enabled LLMs**, which is not covered by prior work. We will revise the presentation to avoid overclaiming and more precisely describe our contributions.
>
> ### Q2. Citation of related work
> We agree that incorporating these works helps better position our research. We will revise the Related Work section to include these references. However, we would like to clarify key differences in our attack scenarios and objectives:
> 1. **Operationalizing a Threat Model for Red-Teaming LLMs (Verma et al., 2025).**
> While our work shares similarities with Infusion Attacks, those attacks typically assume the adversary can manipulate in-context data (like the retrieved documents). In contrast, CREST-Search is strictly limited to manipulating the user input query under a complete black-box constraint.
>
>
> 2. **Rainbow Teaming (Samvelyan et al., 2024).**
> Rainbow Teaming uses open-ended search for diverse prompts. CREST-Search uses a **strategy-driven design** that targets specific failure modes in the citation pipeline. Our experiments show that optimizing within these structured strategies leads to more effective exploitation of search-specific risks. Please refer to *Review-Mnsz, Q4* for detailed experimental results.
>
>
> 3. **RedTeamCUA.**
> Although RedTeamCUA operates in web environments, the threat model differs. It assumes the ability to manipulate the **external environment** (e.g., injecting malicious prompts into web pages), whereas our work considers that the attacker can only control the **user input query**.
>
> ### Q3. Baselines
>
> To address this concern, we have additionally included SafeSearch for comparison. The experimental results show that CREST-Search achieves stronger performance in uncovering citation risks.
>
> |Method|Detection Rate|
> |-|-|
> |SafeSearch|38.3%|
> |CREST-Search|80.5%|
>
> The two approaches differ in their threat models. In CREST-Search, the attacker can only manipulate the user query under an online web environment, while SafeSearch evaluates under an offline setting, where the attacker can manipulate retrieved pages. We argue that our setting more closely reflects real-world auditing scenarios.
>
>
> ### Q4. Discussion of defense
>
> We will comprehensively discuss other mitigation methods and their impacts on the safety, efficiency, system usability, and deployment in real-world systems.
>
> We will incorporate more detailed experiments and analyses discussed above in the revised version and carefully improve the presentation.

---

> > ### Author Rebuttal · Reviewer_LKaP · 2026-04-04
> >
> > My concerns have been adequately addressed and I have raised my score.

---

> > > ### Author Response · Authors · 2026-04-07
> > >
> > > We sincerely thank the reviewer for the positive feedback and for raising the score. We are glad that our responses have addressed the concerns.

---

### Official Review · Reviewer_Mnsz · 2026-03-13

**Soundness:** 3
**Presentation:** 3
**Significance:** 3
**Originality:** 3
**Overall Recommendation:** 4
**Confidence:** 5

**Summary:**

Unlike prior work, which has primarily focused on unsafe responses generated by web-enabled LLMs, this study shifts attention to unsafe citation risks. It proposes a black-box red-teaming framework with three attack strategies to comprehensively evaluate the safety vulnerabilities of LLMs with web search. In addition, the study introduces WebSearch-Harm, a dataset designed for red-teaming-oriented model fine-tuning. Comprehensive experiments on four commercial web-enabled LLMs show that the proposed approach outperforms existing baselines.

**Compliance With Llm Reviewing Policy:**

Affirmed.

**Final Justification:**

The author's rebuttal has addressed my primary concerns. I believe it is reasonable to maintain the original score of 4.

**Key Questions For Authors:**

- The evaluation does not report how many citations are typically generated per response / how many of them are harmful. I would have liked to see some evaluations on this, since the number of citations included in a response itself could be an important factor.

- Other datasets have a higher proportion of queries flagged by the query toxicity detector, which raises the question of whether the lower risk detection rate of them is due to the model’s safety guard filtering such queries before they can be processed. In addition, datasets such as Red-Eval and DangerousQA appear to show substantially higher citation risk, so it is unclear whether the observed effect of the proposed dataset is driven mainly by jailbreak effectiveness. Maybe it's possible that current models may already have adapted defenses against baselines?

- It would be helpful to quantify how much the refinement rounds increase citation risks and response risks.

- It would also be useful to clarify the performance improvement obtained by combining multiple generation strategies.

- In Table 1, the column names “Detection rate” and “Toxicity” are somewhat ambiguous. Can you clarify this by indicating what each metric refers to explicitly? Also, in Figures 3 and 4, the x-axis tick labels appear too far left-aligned, which makes the figures harder to interpret.

**Limitations:**

Yes

**Strengths And Weaknesses:**

Strengths
- The paper addresses citation risk in web-enabled LLMs, which is a relatively underexplored but important problem.
- The proposed dataset appears to be effective and demonstrates strong performance.

Weaknesses
- The discussion of the evaluation results is somewhat limited, and I would have liked a more detailed analysis.

---

> ### Author Rebuttal · Authors · 2026-03-31
>
> Thanks for the reviewer's valuable suggestions and insightful comments.
>
> ### Q1. Analysis of the number of citations per response and how many of them are harmful
>
> We quantified citation usage and its associated risks as below:
>
> - **Responses w/ Citations (%)**: The percentage of model responses that include at least one citation
> - **Avg #Citations**: The average number of citations, computed over responses that contain at least one citation
> - **Avg #Harmful Citations (w/ Citation Risk)**: The average number of harmful links found in responses that were flagged for citation risk.
>
> |Model|Responses w/ Citations(%)|Avg #Citations|Avg #Harmful Citations(w/Risk)|
> |-|-|-|-|
> |GPT-4o-search|95.87%|5.26|1.46|
> |GPT-4o-mini-search|91.18%|5.22|1.50|
> |Gemini 2.0 search|89.63%|11.09|1.61|
> |Gemini 2.5 search|98.18%|13.67|1.72|
>
> **Key Findings**:
> - Over 89% of responses include citations, confirming that citations are a core feature of these systems.
> - When a risk is detected, typically 1.5 to 1.7 citations are harmful, showing that a single query can lead to multiple unsafe sources.
>
>
> ### Q2. Experiment results analysis of baselines
>
> We would like to clarify that the lower risk detection rate of baselines is not primarily caused by safety filtering, but by their inability to trigger the retrieval-citation pipeline. Our analysis shows that 69.4% of baseline queries receive valid responses. Even when not filtered, these baseline responses rarely include harmful citations.
>
> Moreover, the improved performance of CREST-Search is not due to stronger jailbreak effectiveness. CREST-Search achieves a higher risk detection rate (80.5%) while maintaining lower query toxicity (23.6%). This indicates that our success does not come from bypassing existing safety defenses with new or stronger jailbreak techniques, but from crafting "benign-looking" triggers that exploit retrieval logic. This differs from prior baselines, which primarily aim to trigger unsafe generation and therefore cannot effectively capture this underexplored vulnerability.
>
>
> ### Q3. How do refinement rounds quantitatively affect citation risk and response risk?
>
> We conduct an additional analysis by varying the number of refinement rounds and measuring both citation risk and response risk. The results are summarized below:
>
>
> |Rounds|Citation Risk(%)|Response Risk(%)|
> |-|-|-|
> |1|34.2%|12.7%|
> |3|54.3%|14.3%|
> |5|63.1%|17.2%|
> |7|67.6%|18.9%|
> |10|71.9%|20.4%|
>
> Key observation:
> - Citation risk increases significantly from 34.2% to 71.9%, growing much faster than response risk.
> - This gap suggests that the retrieval-citation pipeline is more vulnerable to iterative optimization than the model's internal text generation safety filters.
> - The framework successfully identifies queries that lead to harmful external content while keeping the model's direct response relatively safe (only 20.4% risk at 10 rounds).
>
> ### Q4. Performance of combining multiple generation strategies
>
> Our experiments show that a simple combination of multiple generation strategies does not outperform optimizing with a single fixed strategy.
>
> We conduct an additional experiment on a victim model (gemini-2.5-flash search) under the "harassment" harmful category. We evaluate a multi-strategy variant where the three generation strategies are applied in a cyclic manner across refinement rounds (e.g., Key Words Injection → Exaggeration → Role Play → Key Words Injection → ...). Other settings are kept the same. The results are summarized below:
>
> |Strategy|Detection Rate(%)|
> |-|-|
> |Key Words Injection|87.1%|
> |Exaggeration|73.5%|
> |Role Play|79.3%|
> |Combined|69.2%|
>
> **Analysis.** The combined strategy yields slightly lower performance compared to the single-strategy setting. Although the combined strategy may introduce greater diversity by exploring multiple attack patterns, it limits its ability to fully optimize and exploit a specific attack direction. In contrast, consistently applying a single strategy benefits from deeper and more focused optimization.
>
> **Discussion.** We agree with the reviewer that combining multiple strategies is a promising direction. Designing mechanisms for attack path exploration and strategy scheduling is an important direction for future work.
>
> ### Q5: Clarification of Terminology
>
> We apologize for the ambiguity in the current version.
>
> - **Detection rate** refers to the proportion of cases where the model output (or its cited content) is identified as risky according to our evaluation framework
> - **Toxicity** refers to the proportion of input queries that are classified as toxic by the toxicity detection model
> - **Figure 3** compares how different baselines expose various types of risks. Its **x-axis** represents different baselines
> - **Figure 4** evaluates the transferability of these risks across different victim models. Its **x-axis** represents different victim models
>
> We will incorporate more detailed analyses above in the revised version and improve the presentation.

---

> > ### Author Rebuttal · Reviewer_Mnsz · 2026-04-03
> >
> > Thanks to the authors for addressing my concerns. I noticed that the average number of citations per query is fairly high, especially for Gemini Search, whereas the average number of harmful citations remains comparatively low. While I acknowledge the importance of framing this as a new threat, I am still not convinced that the proposed system demonstrates sufficiently meaningful practical impact. In particular, when only one citation out of more than ten is harmful on average, the practical severity of the issue does not yet feel fully convincing.
> >
> > In this regard, I believe that the average ratio of harmful citations to total citations per query is an important metric and should be reported. It would also be useful to analyze how this ratio changes as the number of refinement rounds increases. For these reasons, I will maintain my current score.

---

> > > ### Author Response · Authors · 2026-04-07
> > >
> > > We thank the reviewer for this valuable suggestion.
> > >
> > > ### Q1. Practical impact of citation risk
> > >
> > > We would like to clarify that the observed citation risk demonstrates meaningful practical impact. The reasons are as follows:
> > >
> > > - In real-world usage, users do not uniformly evaluate all cited sources. Instead, exposure to even a single harmful citation can be sufficient to mislead users.
> > > - This risk is amplified because citations are presented as supporting evidence for the model’s response, making it more credible. Therefore, even a small number of harmful citations can have a disproportionate impact.
> > > - The AI-related regulations, such as the EU AI Act, also emphasize preventing misleading or harmful information in AI-generated outputs.
> > >
> > > ### Q2. Harmful citation ratio and its progression with refinement
> > >
> > > Following your suggestions, we report *the average ratio of harmful citations to total citations per query* and *how this ratio changes as the number of refinement rounds increases* across four victim models.
> > >
> > > **Table 1: Average harmful citation ratio across different victim models.**
> > >
> > > |Model|Avg #Citations|Avg #Harmful Citations|Avg Harmful Citation Ratio|
> > > |-|-|-|-|
> > > |GPT-4o-search|5.26|1.46|27.76%|
> > > |GPT-4o-mini-search|5.22|1.50|28.74%|
> > > |Gemini 2.0 search|11.09|1.61|14.52%|
> > > |Gemini 2.5 search|13.67|1.72|12.58%|
> > >
> > > **Table 2: Change in harmful citation ratio with refinement rounds across different victim models.**
> > >
> > > |Rounds|GPT-4o-search|GPT-4o-mini-search|Gemini 2.0 search|Gemini 2.5 search|
> > > |-|-|-|-|-|
> > > |1|9.51%|9.05%|7.83%|7.52%|
> > > |3|15.14%|16.45%|10.81%|10.39%|
> > > |5|21.78%|22.77% |12.26%|10.87%|
> > > |7|24.32%|24.79% |13.21%|11.24%|
> > > |10|27.76%|28.74%|14.52%|12.58%|
> > >
> > > The harmful citation ratio ranges from 12.6% to 28.7% across all models. The ratio increases steadily with refinement rounds, indicating that it becomes increasingly effective at steering retrieval toward harmful sources. As a result, the likelihood of exposing users to harmful content becomes substantial and grows with attack strength.
> > >
> > > We will include these analyses and further discussion in the revision to better quantify the practical impact.

---

### Decision · Program_Chairs · 2026-04-30

**Decision:**

Accept (regular)

**Comment:**

This paper introduces CREST-Search, a red-teaming framework for evaluating the safety of LLMs augmented with web search capabilities. The framework proposes three adversarial query generation strategies and an iterative refinement loop, along with a fine-tuning dataset (WebSearch-Harm). The paper makes a solid empirical contribution to an important and timely problem. The identification of citation risk as a distinct safety dimension, the demonstration that 74.7% of detected risks are citation-only, and the comprehensive evaluation across multiple commercial systems represent meaningful contributions that the community can build upon. While the methodological novelty of individual strategies is limited, the unified framework and the empirical validation are valuable. The recommended revisions to the paper include: (1) toning down novelty claims regarding individual strategies, (2) fixing all presentation issues (especially the duplicate figure), (3) expanding the defense discussion, and (4) incorporating the additional experiments and analyses provided during rebuttal.